# Hydroelectrolytic and Acid–Base Parameters after 80 to 115 km Endurance Races (Raid Uruguayo) and Their Association with the Comfort Index

**DOI:** 10.3390/ani13040670

**Published:** 2023-02-14

**Authors:** Gonzalo Marichal, Pablo Trigo, Carlos Soto, Ana Meikle, Gonzalo Suárez

**Affiliations:** 1Unidad de Clínica y Cirugía de Equinos, Departamento Hospital y Clínicas Veterinarias, Facultad de Veterinaria, Universidad de la República, Montevideo 11800, Uruguay; 2IGEVET CONICET CC La Plata, Facultad de Ciencias Veterinarias, Universidad Nacional de la Plata, La Plata 1900, Argentina; 3Departamento Hospital y Clínicas Veterinarias, Facultad de Veterinaria, Universidad de la República, Montevideo 11800, Uruguay; 4Laboratorio de Endocrinología y Metabolismo Animal, Facultad de Veterinaria, Universidad de la República, Montevideo 11800, Uruguay; 5Unidad de Farmacología y Terapéutica, Departamento Hospital y Clínicas Veterinarias, Facultad de Veterinaria, Universidad de la República, Montevideo 11800, Uruguay

**Keywords:** equine, weather, hydroelectrolyte concentration, endurance

## Abstract

**Simple Summary:**

The Raid Uruguayo (RAID) is a competitive endurance race (80 to 115 km long) where horses must cover two-thirds of the distance in a single stage. This study is a descriptive analysis of the hydroelectrolytic parameters of 900 horses that participated in RAIDs in a calendar year under different climatic conditions. Hematocrit, total plasma proteins, and blood pH were measured before and after the race. Electrolyte losses in horses that did not finish the race were more pronounced when weather conditions were more severe. The data confirmed the relevance of climate conditions on hydroelectrolytic losses for RAIDs.

**Abstract:**

The Raid Uruguayo (RAID) is an equestrian endurance competition. This study characterized the hydroelectrolytic parameters (Na^+^, K^+^, Cl^−^, tCa^++^, and iCa^++^), hematocrit (Ht), total plasma protein (TPP), and blood pH from 900 equine athletes (finishers and non-finishers) competing over distances of 80 to 115 km under different climate conditions. Paired blood samples were taken prior to the start of the competition (sample 1) and at the end of the race or at the time of leaving the competition (sample 2). The association of the comfort index (CI: low, moderate, and high) with blood parameters was evaluated. Of the 900 horses included, 550 were not able to finish the trial. The comfort index was not associated with success in completing the race. In the horses that finished the race, the CI was not associated with Ht, pH, TPP, or Na^+^ concentrations in samples taken after finishing the RAID. In contrast, the decreases in chloride, K^+^, tCa^++^, and iCa^++^ concentrations found after the race were more pronounced at moderate and high CI values when compared with low CI values. In horses that did not finish the race, the CI was associated with all variables except for Ht. The data confirmed the relevance of considering the impact of the comfort index in hydroelectrolytic losses in the RAID, as it influence ssuccess or failure in the performance of endurance horses finishing the competition.

## 1. Introduction

Endurance races are considered to be competitions with some of the highest metabolic demands and hydroelectrolyte imbalances [1]. Despite all the medical checks, the number of horses that do not complete such races is still high: one of the most comprehensive retrospective studies which included 30,741 endurance horses competing over distances between 100 and 160 km reported that one-half of the competitors were eliminated [2].

The endurance RAIDs performed in Uruguay include distances of 80, 90, 95, and 115 km. Horses must cover two-thirds of the distance in the first phase, and the rest in the second phase. There is one mandatory rest period, with veterinary controls, between both phases (Uruguayan Equestrian Federation, FEU 2021, www.federacionecuestreuruguaya.com.uy (accessed on 30 January 2023). Winning horses average speeds between 24 and 34 km/h. In a 115 km RAID, horses must complete the course in only two phases, whereas in a 120 km FEI endurance race, they usually complete it in five phases.

Prolonged physical exercise induces physiological phenomena of a challenging magnitude; indeed, homeostatic imbalances occur with a higher frequency than in any other equestrian sport [1,2,3,4,5,6,7]. It has been shown that vigorous exercise in hot and humid conditions produces a greater electrolyte imbalance and dehydration than in dry and cool conditions [4,5], which can also generate clinical metabolic problems [6,7]. Di Battista et al. [8] reported that the minimum temperature is a risk factor for elimination due to claudication and metabolic reasons, although the mechanism is not entirely clear. Jones [9] proposed a comfort index (CI) based on both temperature (T) and humidity (H) (CI = T (°F) + H (%)), with three categories (low, moderate, and high). As far as we know, no data are available regarding the association of CI with hydroelectrolytic and hematological alterations in endurance races.

The objective of the present study was to determine whether climatic conditions (CI) affect the hydroelectrolyte and acid–base values in equine athletes after RAID races. Our hypothesis is that the completion of RAIDs and the hydroelectrolytic and acid–base parameters after the race are associated with the comfort index.

## 2. Materials and Methods

### 2.1. Climate Data

The Uruguayan Institute of Meteorology (Inumet, Uruguay) provided temperature (T, in degrees Fahrenheit) and humidity (H, in percentage) data at the beginning of each race.

### 2.2. Comfort Index

The CI was calculated according to the following formula [9]: CI = T (°F) + H (%); 3 CI levels were identified: low (<130 CI), moderate (130–150 CI), and high (>150 CI).

### 2.3. Competences and Animals

Competitions (*n* = 34) between March and November (covering all seasons; however, no races take place in summer) were included at different locations in Uruguay. Competences are considered long when the distances covered are over 80 km (FEU 2021). RAIDs in this study were categorized as distances of 80 km (*n* = 4), 90 km (*n* = 24), and 95 to 115 km (*n* = 6). All equine athletes included in this study (*n* = 900) were registered in the FEU. Breeds included Arab, Creole, Thoroughbred, and crosses. They were at least 5 years old regardless of sex and health and were free of locomotor disabilities, cardiorespiratory, and/or digestive conditions or any other symptom or injury which, in the opinion of the official veterinarians, would endanger the life of the animal. The pre-competition exam is an exam where clinical parameters such as heart rate, respiratory rate, hydration status, intestinal motility, and capillary refill time are checked;a locomotor control is also performed, and the horse is examined for any evidence of claudication. Check-ups are carried out the day before the race between 11 a.m. and 1 p.m. by the same team of veterinarians that will remain until the end of the race. The veterinarians who work on the tests are part of a pool of veterinarians certified by the FEU. During the test, at the break, the veterinarians carry out the same metabolic and locomotor examinations as the previous day to control the health status of the horses and authorize those with parameters close to normal to continue in the race (FEU, 2021).

### 2.4. Blood Sampling

In each of the competitions, the official veterinarians took two blood samples (5 mL, obtained by aseptic jugular puncture) from all the equines (1800 samples). The first sample was taken at the admission control, the day before the competition (approximately 20 h before the start of the race). The second sample was taken when the horse’s participation ended. In horses that were eliminated from the race at the break or that abandoned the race, blood samples were taken immediately. In horses that finished the race, blood samples were taken when the competition ended. Part of the sample was placed in tubes with EDTA (Sanli Medical, China), and the other sample was immediately transferred to tubes with a coagulation accelerator and Serotubpp gel separator (Sanli Medical, China), and centrifuged for 15 min at 3600 rpm (Rolco Model197, Argentina); the serum was transferred to Eppendorf microtubes. These were kept in an isothermal box with ice cubes for transport until refrigeration at 4 °C, where they were stored until further processing in the laboratory, within 48 h after obtaining them. In the blood samples mixed with EDTA, hematocrit (Ht) was determined by the centrifugation of capillaries in a micro centrifuge, and total plasma proteins (TPP) concentrations were estimated by refractometry (Biriden, Uruguay).

### 2.5. Determination of Hydroelectrolyte and Acid–Base Values in the Laboratory

The serum samples were analyzed in the Laboratory of Clinical Analysis of the Veterinary Hospital (Facultad Veterinaria, Universidad de la República). A high-performance specific ion electrode analyzer (HumaLyte plus 5 automatic, Germany) was used to determine serum concentrations of sodium (Na^+^), potassium (K^+^), chloride (Cl^−^), total plasma calcium (tCa^++^), ionized calcium (iCa^++^), and serum pH.

### 2.6. Statistical Analysis

Descriptive analyses of the hydroelectrolyte and acid–base values were carried out with the determination of the main cut-off values (mean and standard deviation (SD); 5th, 25th, 50th, 75th, and 95th percentiles, and minimum/maximum). Statistical differences in race data were calculated using a chi-squared test. Independently for the animals that did or did not finish the race, the effects on each hydroelectrolyte, acid–base, and hematological value were analyzed with linear mixed models, considering the fixed variables (CI, time of sampling and interaction) and random effects (animal variables within competition and distance). *p*-values of less than 0.05 were considered statistically significant. All analyses were performed in R (Version 4.1.2) [10].

### 2.7. Ethical Approval

This project was approved (number 111400-000124-12) by the Honorary Committee of Ethics and Animal Experimentation (CEUA) of the Facultad Veterinaria, Universidad de la República (Uruguay). In addition, this study received approval from the Uruguayan Equestrian Federation, owners, riders, and official and private veterinarians of Ministerio Ganadería Agricultura y Pesca (MGAP, Uruguay). All the procedures performed on the animals had the prior consent of their owners and were carried out under strict biosecurity procedures as well as under the supervision of those responsible for the study.

## 3. Results

### 3.1. General Description of the Data

In the period from March to November, 99 animals participated in 80 km races (*n* = 42 and 57 with moderate and high CI values, respectively), 693 participated in 90 km races (*n* = 278, 238, and 177 with low, moderate, and high CI values, respectively), and 108 participated in >95 km races (*n* = 25, 43, and 40 with low, moderate, and high CI values, respectively). From the 900 horses included in this study, 350 finished, whereas 550 (61%) did not. Of the total number of participants, 34% in 80 km races (*n* = 34), 40% in 90 km races (*n* = 277), and 36% (*n* = 39) in ≥95 km races completed the competitions. There were no significant differences between the proportions of animals that completed the event according to distance (chi-squared test, *p* > 0.05). Table 1 shows the number of horses according to the CI (chi-squared test, *p* = 0.447). No effect of the CI was observed in the proportion of horses that finished the race: 34% of the horses completed races with low CI scores; 36% completed races with moderate CI scores; and 30% of the horses completed races with high CI values (chi-squared test, *p* > 0.05).

### 3.2. Hematocrit, pH, Total Plasma Proteins, and Electrolyte Concentrations of Horses Pre-Competition

Table 2 shows the previous cut-offs of hydroelectrolyte, acid–base, and hematological values of pre-competition equines.

### 3.3. Hematocrit, pH, Total Plasma Proteins, and Electrolyte Concentrations of Horses That Finished the Race

Hematocrit and TPP increased at the end of the race (*p* < 0.05 for both) and was not affected by the CI (*p* > 0.05) (Figure 1A,B). The concentrations of ionic calcium and total calcium decreased at the end of the race (*p* < 0.05), and the interaction with CI was significant (*p* < 0.05); at the end of the race, the concentration of calcium was lower in high CI values than in low CI values (Figure 1C,D).

The concentrations of chloride decreased at the end of the race (*p* < 0.05), depending on the CI (*p* < 0.05), because the decrease in sample 2 was lower in those with low CI values when compared with moderate and high CI values (Figure 2A). The sodium concentration was not affected by the time of sampling (first or second sample), nor by the CI (Figure 2B).

Potassium concentrations were affected by the interaction between samples and CI because the concentration decreased in moderate and high CI values with respect to low CI values (Figure 2C). The pH, unlike the behavior of chloride and potassium, increased at the end of the race (*p* < 0.05) without significantly affecting the CI or the interaction between both (Figure 2D).

### 3.4. Hematocrit, pH, Total Plasma Proteins, and Electrolyte Concentrations of Horses That Did Not Finish the Race

Figure 3 and Figure 4 show the variables of the horses that did not complete the event. Sample 1 was taken before the race in all animals, whereas Sample 2 was obtained at times when horses were eliminated or had left the race; thus, there was no uniformity in the time of sampling in relation to the initiation of the race.

The hematocrit increased at the end of the race (*p* < 0.05), and it was not affected by the CI or by an interaction between the two (Figure 3A). The total plasma protein concentrations increased at the end of the race (*p* < 0.05) and were affected by the CI and by the interaction between these two factors, showing differences in total plasma protein concentrations in Sample 2 between low and high CI (Figure 3B). The concentrations of total and ionic calcium decreased at the end of the race (*p* < 0.05), and the interaction between CI and the time of sampling was significant (*p* < 0.05); in sample 2, it was lower for high CI values than for low CI values (Figure 3C,D).

Chloride concentrations decreased at the end of the race (*p* < 0.05), and the interaction between CI and the time of sampling was significant (*p* < 0.05); the decrease was lower for low CI scores than in those with moderate and high CI values (Figure 4A). The sodium concentration was not affected by the time of sampling, and it was lower in moderate and high CI relative to low CI (Figure 4B). Potassium concentrations were also affected by the interaction between CI and the time of sampling. The potassium concentration in Sample 2 was lower in moderate and high CI values than in low CI values (Figure 4C). The pH, independently of the CI, increased at the end of the race (*p* < 0.05), and was lower in high CI compared with low CI (Figure 4D).

## 4. Discussion

This is the first study to investigate the association between an indicator based on climatic conditions (comfort index) and hydroelectrolyte concentrations, total plasma proteins, hematocrit, and pH of equine RAID athletes that did or did not complete the race. This was a descriptive analysis performed under field conditions, where many variables, such as sample population and variability associated with horse background and the environment of the races, could not be controlled. This limitation should be considered in the interpretation of the results.

Although RAID is a competitive sport, the primary goal for most competitors is the completion of the race; the main causes of horse retirement are lameness in the horse or perceptions of fatigue by the rider. In our study, 61% of the horses failed to finish the race. Nagy et al. [11] found that 54% of 4326 horses that ran more than 100 km did not finish the race. The same research group [2] reported that 51.4% (739 of 1435) of the horses registered in races of 80–160 km did not complete the race. Younes et al. [12] mentioned that 38.9% (*n* = 2738) of the horses that participated in competitions with the same distance range of the previous study were eliminated. This data was similar to other studies that reported a global elimination rate of 38.7 to 45% [13], but higher than the 18.9 and 31.4% reported by other authors [1,14,15].

The great variability in the completion ranges has been associated with the location of the races, distance, number of competitors, climatic conditions, or characteristics of the terrain and altitude [16,17]. In addition, factors such as riding strategy, speed, and age are related to the elimination rate [18]. In our study, the elimination ranges were notably higher than in other reports, which may be due to the lack of preparation of the animals and/or that, in the RAID, the horses must complete the race in only two stages (covering two-thirds of the distance in the first stage). One of the limitations of the present study is the lack of information regarding the reason for eliminating the horse from the race. In contrast to the international literature, which has shown that vigorous exercise in hot and humid conditions is associated with clinical metabolic problems [6,7,19], in our study, no association of the comfort index (CI) was found in relation to the proportion of horses that did or did not finish the races. The low relevance of the CI in the completion of or retirement from the races may be associated with the fact that there are no competitions in summer, where high temperatures could have a greater impact on the non-completion of horses in this type of race.

As reported previously [20], hematocrit was higher at the end of the race. In our study, this increase reached up to 10% more than the baseline value and was not associated with the CI. Djoković et al. [17] described an increase in Ht of 9.6% at the end of the race. This hemoconcentration is known to be a result of splenocontraction and a reduction in plasma volume due to fluid lost due to sweating [1,21,22,23]. The increase in the concentration of total plasma proteins at the end of the race was consistent with previous reports [24,25]. Animals in this study exhibited degrees of dehydration which, according to the literature, varied from moderate (50% Ht and 8 g/dL total plasma proteins) to severe or possible shock (>50% Ht and >8 g/dL total plasma proteins) [15,18,26]. In horses that did not finish the race, total protein plasma concentrations were greater in high vs. low CI values.

The sodium concentrations were not affected by the time of sampling, which agrees with the findings of Flaminio and Rush [6], who reported that Na+ levels remain relatively normal. On the other hand, several authors [7,20,25,27,28] have reported hyponatremia at the end of endurance races. Here, sodium concentrations were not associated with CI in the horses that finished; however, in non-finishing horses, it was lower in those with high CI vs. low CI values. This is consistent with the total plasma protein concentrations in these groups and may reflect poorer hydroelectrolytic conditions in horses that do not finish races under challenging weather conditions.

Hypochloremia identified at the end of the races corroborated the literature on endurance [20,28]. The chloride decrease was less in low CI than in moderate and high CI. Schott et al. [27] reported similar findings regarding hypochloremia in horses that ran 160 km, with no differences between those that finished and those that did not. The latter findings differ from those of Fielding et al. [1], who reported that the horses eliminated in endurance races had a lower concentration of chloride compared with those that finished the race. It is known that losses of chloride through sweat are important [22]; in our study, the decrease in this ion was important (up to 15% with respect to pre-race values).

Regarding the K^+^ concentration in the equines that finished the race, although no differences were found before and after the end of the race in low CI values, the concentration decreased at the end of the race in those with moderate and high CI scores. These results are expected, given the adverse conditions and the higher sweating expected in moderate and high CI with respect to low CI. Intense sweating (4% to 7% of the equine body weight) can justify these losses, because sweat has a hypertonic composition [5,17]. An unexpected finding was the increase in K^+^ in horses that did not finish the race in low CI values, whereas it decreased in those with moderate and high CI scores. At present, we posit an obvious explanation for these results.

Blood pH increased after the race, although only in the horses that did not finish; a lower pH was found in high CI compared with low CI. Metabolic alkalosis occurring in endurance horses is secondary to massive chloride loss in sweat, compensatory renal bicarbonate reabsorption, and the excretion of hydrogen in exchange for sodium. In addition, metabolic alkalosis can develop in hyperthermic horses as a result of pulmonary hyperventilation [29,30,31,32].

In this study, the concentrations of iCa^++^ and tCa^++^ decreased at the end of the race, in agreement with Bernardi et al. [20]. In addition, concentrations of both types of calcium were lower in high CI compared with low CI for both finishing and eliminated horses. As mentioned above, sweat losses of sodium, chloride, potassium, and, in part, calcium and magnesium, when added to metabolic alkalosis, are the genesis of diaphragmatic synchronous flutters [33]. McCutcheon and Geor [34] mentioned that the total losses of ions and fluids through sweat in moderate-intensity exercise training in hot and humid environmental conditions were more than double those that occur during training in cool and dry conditions.

The relevant changes found at the end of the race and in the large proportion of horses not finishing the competitions is indicative of the high physical demand that this endurance sport requires of horses. Although there were differences in race distances, the number of phases, race speeds, and elimination rates, hydroelectrolytic disturbances in RAIDS are comparable to FEI endurance races. As a recommendation, longer recovery periods should be considered in tests that will take place on days with moderate and high CI.

## 5. Conclusions

In conclusion, hematocrit, total plasma proteins, pH, and Na^+^ were not associated with the comfort index in horses that finished the race, although the other electrolytes assayed (iCa^++^, tCa^++^, Cl^−^, and K^+^) were. In horses that did not finish the races, all variables except hematocrit were associated with the comfort index. Thus, we concluded that environmental conditions had a greater impact on the concentrations of electrolytes in horses that did not complete the event compared with those that did and supports the concept of the impact of the comfort index as a risk factor in RAID.

## Figures and Tables

**Figure 1 animals-13-00670-f001:**
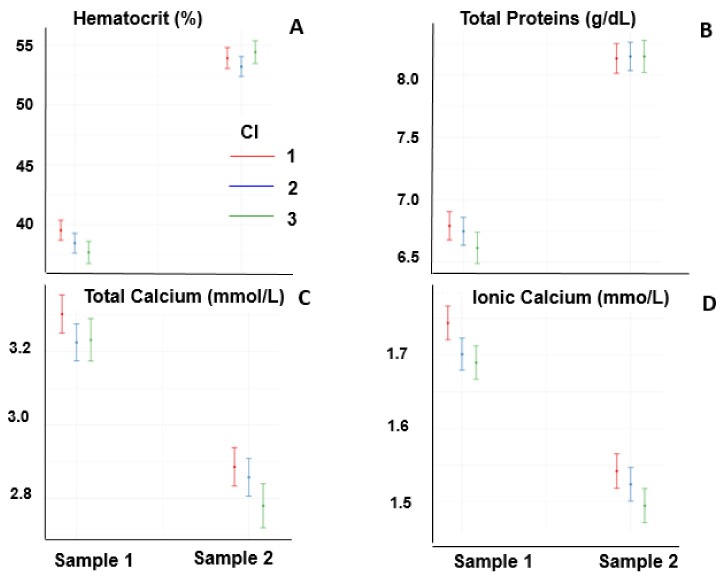
Hematocrit (**A**), total plasma protein (**B**), total calcium (**C**), and ionic calcium (**D**) of horses one day before the race (Sample 1) and at the end of race (Sample 2).

**Figure 2 animals-13-00670-f002:**
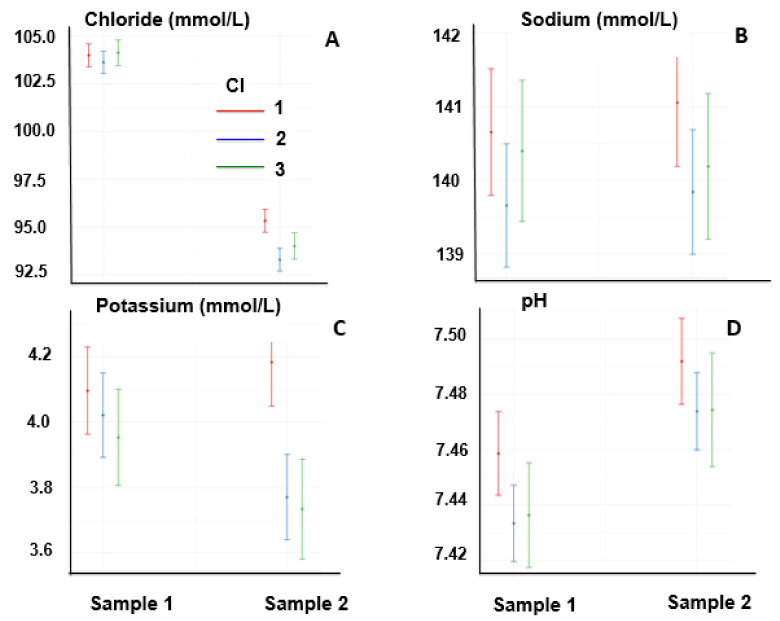
Chloride (**A**), sodium (**B**), and potassium (**C**) concentrations, and pH (**D**) of horses one day before the race (Sample 1) and at the end of the race (Sample 2).

**Figure 3 animals-13-00670-f003:**
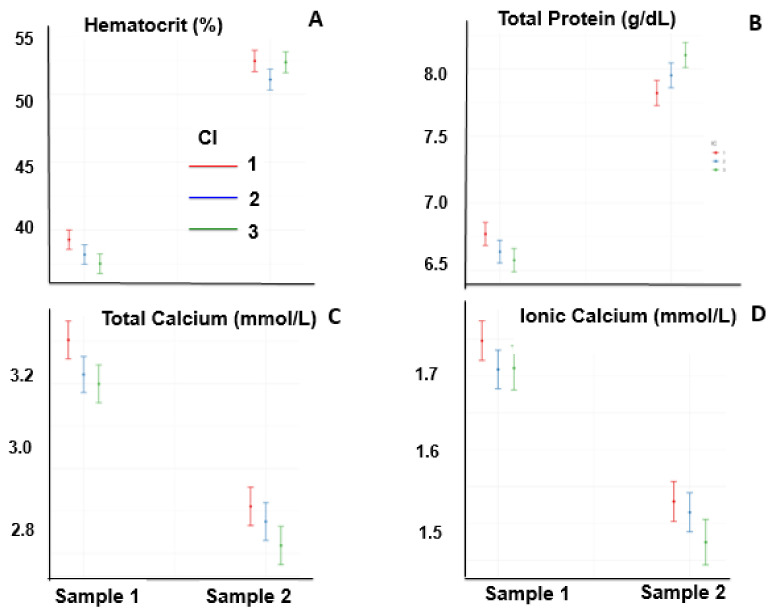
Hematocrit (**A**), total plasma protein (**B**), total calcium (**C**), and ionic calcium (**D**) of horses one day before the race (Sample 1) and at elimination or when they retired from the race (Sample 2).

**Figure 4 animals-13-00670-f004:**
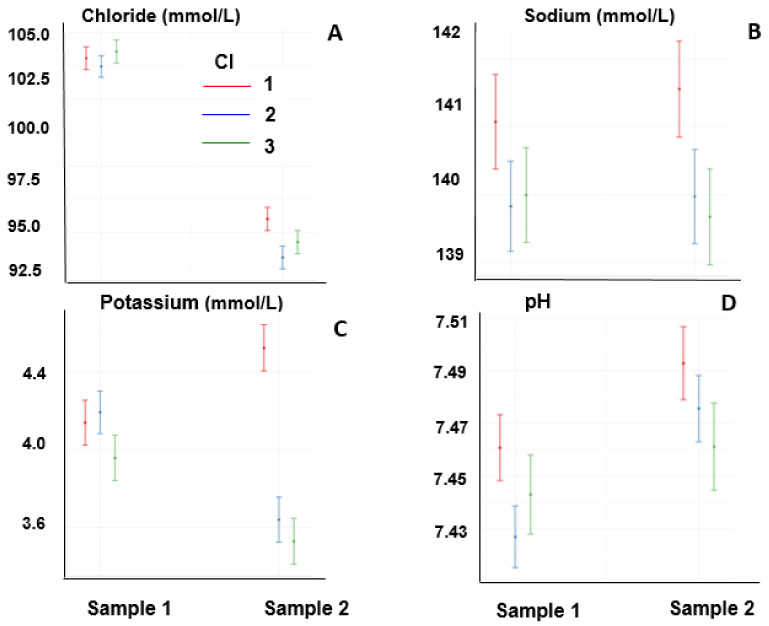
Chloride (**A**), sodium (**B**), and potassium (**C**) concentrations, and pH (**D**) of horses one day before the race (Sample 1) and at elimination or when they retired from the race (Sample 2).

**Table 1 animals-13-00670-t001:** Number of horses that participated in the races according to the comfort index (CI) in 34 competitions carried out between March and November across different locations in Uruguay.

Comfort Index	Status in the Competition		Total
	Finishers	Non-Finishers	
Low (<130)	123	180	303
Moderate (130–150)	129	194	323
High (>150)	98	176	274

**Table 2 animals-13-00670-t002:** Hydroelectrolyte, acid–base, and hematological values for horses pre-competition.

Variable	*n*	
Hematocrit	876	(%)
Average (SD)		38.5 (3.7)
Median (5%, IQR, 95%)		38.0 (33.0, 36.0, 40.0, 45.0)
Minimum:Maximum		28.0:64.0
Total Plasma Protein	876	(g/dL)
Average (SD)		6.69 (0.40)
Median (5%, IQR, 95%)		6.70 (6.00, 6.40, 7.00, 7.32)
Minimum:Maximum		5.60:7.60
K^+^	886	(mmol/L)
Average (SD)		4.07 (0.56)
Median (5%, IQR, 95%)		4.05 (3.20, 3.70, 4.40, 5.00)
Minimum:Maximum		2.60:7.30
Na^+^	887	(mmol/L)
Average (SD)		140 (4.2)
Median (5%, IQR, 95%)		141 (133, 139, 143, 145)
Minimum:Maximum		119:160
Cl^−^	887	(mmol/L)
Average (SD)		104 (2.5)
Median (5%, IQR, 95%)		104 (100, 103, 105, 107)
Minimum:Maximum		89:121
iCa^++^	886	(mmol/L)
Average (SD)		1.67 (0.11)
Median (5%, IQR, 95%)		1.69 (1.50, 1.60, 1.72, 1.80)
Minimum:Maximum		0.80:1.90
tCa^++^	886	(mmol/L)
Average (SD)		3.25 (0.22)
Median (5%, IQR, 95%)		3.28 (2.98, 3.16, 3.38, 3.50)
Minimum:Maximum		1.50:3.70
pH	676	
Average (SD)		7.55 (0.08)
Median (5%, IQR, 95%)		7.56 (7.40, 7.50, 7.60, 7.66)
Minimum:Maximum		7.20:7.72

## Data Availability

The datasets used and analysed during the current study are available from the corresponding author on reasonable request.

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
