# Peer review of "Hydroelectrolytic and Acid–Base Parameters after 80 to 115 km Endurance Races (Raid Uruguayo) and Their Association with the Comfort Index"

_animals, 2023, doi:10.3390/ani13040670_

Round 1

Reviewer 1 Report (New Reviewer)

1. what the comfort index is and how it is determined along with the various categories should be discussed earlier in the paper (in the introduction rather than waiting until the methods section).

2. What were the horses weights and body condition scores? This could have an impact on ability to complete the race.

3. Diet can have some impact but understand this would be almost impossible to evaluate in the current study design.

4. The results are not surprising.  Changes in calcium uptake can be affected by conditioning.  

5. Sweat loss will affect several of the electrolytes and while the authors mention this effect there are connections again to diet here which can affect the strong ion difference (SID) which can affect fatigue and the ability to finish.

6.  The fact that there was no effect of comfort index on those horses that did complete the race does suggest that conditioning has an effect on the ability to complete the race.  The fact that comfort index did affect those that did not finish the race suggests that as the authors state that they may be more susceptible to environmental conditions in the moderate or high comfort indexes.

7. While field survey studies can be valuable as this study is, it is important to remember that this is not a controlled study and there is more going on here rather than just the change in comfort index.

Author Response

We are now sending the revised version that we hope fulfills with the requirements of Animals. We acknowledge the reviewers’ criticism, and we state all corrections and comments below; changes in the new manuscript are highlighted. To be clearer, we have kept all comments of the reviewers and in bold italics our response.

Although the English grammar and spelling have been improved by the authors, none of us is a native English speaker, and therefore some errors are beyond our control. Therefore, in conjunction with the submission of this review, we are submitting the article for English review according to the journal's recommendations.

Reviewer 1

  1. What the comfort index is and how it is determined along with the various categories should be discussed earlier in the paper (in the introduction rather than waiting until the methods section).

Au: Corrected. The formula has been added in the introduction: “Jones [9] has proposed a Comfort Index (CI) that is based on both temperature (T) and humidity (H) (CI = T (°F) + H (%)), with three categories (low/moderate/high).

  1. What were the horses' weights and body condition scores? This could have an impact on ability to complete the race.

Au: We completely agree with the reviewer. Unfortunately, this type of data was not registered.

  1. Diet can have some impact but understand this would be almost impossible to evaluate in the current study design,

Au: Again, we agree, and we are considering a study with body conditions and diet in the near future, but this was not included in this study.

  1. The results are not surprising.  Changes in calcium uptake can be affected by conditioning.  

Au: Most results were expected, we believe that data regarding horses that did not finish the competences is novel.

  1. Sweat loss will affect several of the electrolytes and while the authors mention this effect there are connections again to diet here which can affect the strong ion difference (SID) which can affect fatigue and the ability to finish.

Au: We completely agree with the reviewer. As described in the paper the 900 horses were competing in 34 races for several months and we do not have individual data.

  1. The fact that there was no effect of comfort index on those horses that did complete the race does suggest that conditioning has an effect on the ability to complete the race.  The fact that comfort index did affect those that did not finish the race suggests that as the authors state that they may be more susceptible to environmental conditions in the moderate or high comfort indexes.

Au: In the horses did complete the race, the comfort index affected total and ionic calcium chloride and potassium concentration, but not other parameters (hematocrit, total proteins, pH and sodium). In contrast, in horses that did not complete the race all parameters -except hematocrit -were affected by the comfort index. We agree with the reviewer reasoning.

  1. While field survey studies can be valuable as this study is, it is important to remember that this is not a controlled study and there is more going on here rather than just the change in comfort index.

Au: We completely agree with the reviewer. As stated in the discussion: “The great variability in the completion ranges in the literature is related to the location of the races, distance, number of competitors, climatic conditions or characteristics of the terrain and altitude [16, 17]. In addition, factors such as riding strategy, speed and age (≤ 8 and ≥ 15 years) are also related to the higher elimination rate [18]. The fact that, in our study, the elimination ranges were rather higher than other reports may be due to the lack of preparation of the animals and that in RAID, the horses must complete the race in only two stages (covering 2/3 of the distance in the first stage)”.

We have now included in the first paragraph of the discussion section the following phrase that is suggested by the reviewer that may help the reader to be conscious of the limitation of our study: “This is a descriptive analysis performed under field conditions where many variables as sample population and variability associated with horse background and the environment of the races could not be controlled. This limitation should be considered for the interpretation of the results.”

Reviewer 2 Report (New Reviewer)

General Comments:  The authors repeatedly state that CI had or did not have an effect.  The study is a descriptive study not a cause and effect study or a mechanistic experiment with a wide variety of factors that affected the CI.  The authors need to revise all of the statements to read that there was no association or an association between the CI and the various factors measured.  Effect vs. association!!!!!!!!! The discussion is too long and overly speculative in many cases.

There is no control group.  How do we know that the horses were in a steady state before they competed?  There should have been a minimum of two samples taken before the competitions.

How many horses were over 16 years of age?

Line 300: speculation no proof.

Line 329:  back up the statement with a referene.

Author Response

We are now sending the revised version that we hope fulfills with the requirements of Animals. We acknowledge the reviewers’ criticism, and we state all corrections and comments below. To be clearer, we have kept all comments of the reviewers and in bold italics our response.

Although the English grammar and spelling has been improved by the authors, none of us is a native English speaker, and therefore some errors are beyond our control. Therefore, in conjunction with the submission of this review, we are submitting the article for English review according to the journal's recommendations.

 Reviewer 2

General Comments:  The authors repeatedly state that CI had or did not have an effect.  The study is a descriptive study not a cause and effect study or a mechanistic experiment with a wide variety of factors that affected the CI.  The authors need to revise all of the statements to read that there was no association or an association between the CI and the various factors measured.  Effect vs. association!!!!!!!!! The discussion is too long and overly speculative in many cases.

Au: We completely agree with the reviewer, and it was our intention to stablish only the association as it is indicated in the title of the manuscript. On the other hand, the statistical analysis to test this association was to include the CI as fixed effect, and when writing results, we have no other alternative than to state that was significant or not. In discussion section, we changed to expression to association, we have eliminated speculations (see below) and we have shortened this section by 20%.

There is no control group.  How do we know that the horses were in a steady state before they competed?  There should have been a minimum of two samples taken before the competitions.

Au: This was a prospective field study and considering the type of management of the competencies it was not possible to take more blood samples. This was not an experimental design in which we could control the conditions.  We agree with the reviewer, and we are now designing a proposal with fewer animals with a better description of the steady state (including more blood samples, time and type of training).

How many horses were over 16 years of age?

Au: Unfortunately, this type of data was not registered

Line 300: speculation no proof.

Au: We have now changed the expression: The great variability in the completion ranges in the literature is related to the location of 300 the races, distance, number of competitors, climatic conditions or characteristics of the 301 terrain and altitude [16, 17], now reads “The great variability in the completion ranges has been related to the location of the races, distance, number of competitors, climatic conditions or characteristics of the terrain and altitude [16, 17].”

Line 329:  back up the statement with a reference.

Au: We completely agree with the reviewer and have eliminated the two phrases that contained these speculations: “The CI did not affect total proteins in finishers, but in horses that were eliminated, total protein concentrations ​​were higher in High CI in relation to Low CI, indicating that environmental conditions affect the degree of dehydration. The finding suggests that the degree of dehydration in these animals may have been more important, leading the horses to abandon or be eliminated from the race”.

Reviewer 3 Report (New Reviewer)

First off, authors should be commended for doing research on endurance horses within their competitive environment, rather than in a research-based laboratory, as it simulates the strain and stress of what the equine athlete experiences firsthand. Nevertheless, it’s important to note that in doing this real-life type of research setting, it introduces variables that cannot be controlled, and in turn, adds a spectrum of unknown variables to the results that are measured. It also limits the use of controls making it difficult to rule out potential unsupported conclusions. This among other shortfalls within this manuscript stress the importance of proceeding with caution when evaluating the impact of this research to the horse industry.

As for specific areas to address, starting with the title, the “900” does not need to be included within the title. Size of sample population within a title is not needed and it is deceptive as there were not 900 animals within one type of race evaluated. In fact, the sample population within the three race lengths was quite unbalanced with a significant drop within the shorter race compared to the other race types. It may be more useful to indicate within the title the variation of races sampled. The abstract also does not address the 900 came from multiple types of race distances and that the study compared these various race types. These various race types appears important within the results, and thus, should be a part of both the title and the abstract.

The introduction lacks substantial discussion of the physiological impact of endurance races on the horse, and yet, there is much work that has been done previously in this area, so that these references need to be included within the introduction. If publications specific to hydro electrolyte parameters, hematocrit, total proteins, and blood pH of equine endurance competitors are limited, then, discuss what has been done in other equine competitors or even what has been done in human marathon and/or triathlon competitors. The impact of this type of competition on the athlete, even outside of endurance and/or outside of the equine athlete, would be useful in understanding the value of the work being done within this study. In addition, much of the focus of the introduction and even within the discussion is on not finishing the endurance race within the equine athlete, but how is not finishing a race consider a bad thing? Address maybe the potential financial loss and/or the associated health risks that are involved with an athlete that cannot finish a race. Numbers of non-finishers does not explain why this research is important to those less familiar of the sport. Could not finishing an endurance race be associated with the rider not able to finish or behavioral issues with the horse that makes it hard to finish rather than reasons associated with the variables measured within this study? Could additional reasons include health problems not tested in this study such as lameness? Maybe research within the introduction supporting that non-finishing in endurance is primarily due to what was measured within this study would help to emphasize the value of this research. It would also be helpful to address within the introduction how alterations with hydro electrolyte and hematocrit parameters within the equine athlete can have a long-term negative impact on these animals. If research is lacking, a discussion even within the human athlete could be utilized to relate to potential issues within the equine athlete.

Along with justifying what was studied and why it was studied, a hypothesis should be given after the objective statement within the introduction. In addition, while the authors add in a justification statement after the objective statement by saying that this research will “contribute to animal welfare and disease prevention”, these areas were not measured within this study. Welfare is usually measured within more behavioral based measurements using such markers as cortisol and vital signs, but this was not done within this study. In addition, no disease detection was given within this study and authors do not explain how disease relates to the measurements given. Authors need to stay focused on what was truly measured and it’s direct impact on the equine endurance athlete when including a justification.

As for the methods section, more details need to be given concerning the 900 horses sampled. Did each horse only do one race or was there a potential that a horse was sampled for more than one race for this study? Age, height, weight, breed, gender, amount of training, and which type of race attempted needs to be given for the horses sampled for this study as these variables can impact results. These variables can be reported utilizing means (SD) or ranges for the 900. This information would be useful given in a table format to visualize the differences between the three types of race distances reported. Furthermore, for determining the health of the horses utilized for this study, how thorough of a clinical examination was done for each horse? Did it include diagnostic testing utilizing radiographs, endoscopes, and ultrasound? Was a clinical history taken to determine any previous health issues tracked by a veterinarian? Was a full lameness evaluation done with flexion tests utilizing nerve blocks and hoof testers? Did the same veterinarian perform the health evaluations for all horses? How soon before the race were these health evaluations performed? What examinations were done during the race and was it done by the same veterinarian as the initial examination? How soon after was the blood samples taken at the end of the race and was it consistent for all?

As for the discussion, this section begins with a focus on whether horses finished the race, but it does not appear that any documentation was made as to why the horses did not finish the race. Which horses were specifically pulled by the veterinarian inspecting the horses during the race, and of those pulled by the veterinarian, what specifically was the reasons documented for why? What was noted on their records such as an increased vital sign or signs of dehydration? While alterations may have been noted in the measurements recorded for this study for these animals, those alterations may not have stopped a competitor from finishing the race or may not have been the sole reason why. Without notes from the veterinarian for reasons why a horse was pulled or even from the owner, it is hard to support that the measurements taken by the authors were directly associated to not finishing the race. It is especially difficult to make any conclusions without controls utilized for the study. This would help to eliminate potential unforeseen variables that were not accounted for. This lack of a control along with an unbalanced sample population and variability associated with the horse background and with the environment of the races should be discussed as potential limitations to the study. The discussion section must include limitations to a study so that future studies may work to eliminate these potential issues.

Author Response

We are now sending the revised version that we hope fulfills the requirements of Animals. We acknowledge the reviewers’ criticism, and we state all corrections and comments below. To be clearer, we have kept all comments of the reviewers and in bold italics our response.

Although the English grammar and spelling have been improved by the authors, none of us is a native English speaker, and therefore some errors are beyond our control. Therefore, in conjunction with the submission of this review, we are submitting the article for English review according to the journal's recommendations.

Reviewer 3

First off, authors should be commended for doing research on endurance horses within their competitive environment, rather than in a research-based laboratory, as it simulates the strain and stress of what the equine athlete experiences firsthand. Nevertheless, it’s important to note that in doing this real-life type of research setting, it introduces variables that cannot be controlled, and in turn, adds a spectrum of unknown variables to the results that are measured. It also limits the use of controls making it difficult to rule out potential unsupported conclusions. This among other shortfalls within this manuscript stress the importance of proceeding with caution when evaluating the impact of this research to the horse industry.

Au: We acknowledge this reviewer´s comments. We have revised the discussion section to avoid speculations and as reviewer 2 had similar comments and suggested shortening the section; we have diminished the discussion by 20%.

As for specific areas to address, starting with the title, the “900” does not need to be included within the title. Size of sample population within a title is not needed and it is deceptive as there were not 900 animals within one type of race evaluated. In fact, the sample population within the three race lengths was quite unbalanced with a significant drop within the shorter race compared to the other race types. It may be more useful to indicate within the title the variation of races sampled. The abstract also does not address the 900 came from multiple types of race distances and that the study compared these various race types. These various race types appear important within the results, and thus, should be a part of both the title and the abstract.

Au: Corrected, the number of animals was deleted from the title as the reviewer suggested. We fully agree with the reviewer respect that the number of animals according to the length of the race is unbalanced; the 90 km race is the most frequent in Uruguay. Considering this aspect that the reviewer is pointing out, the title now reads ““Hydroelectrolytic and acid base balance in horses after 80 to 115 endurance races (Raid Uruguayo) and its association with comfort index”. This information was also added in the abstract.

The introduction lacks substantial discussion of the physiological impact of endurance races on the horse, and yet, there is much work that has been done previously in this area, so that these references need to be included within the introduction. If publications specific to hydro electrolyte parameters, hematocrit, total proteins, and blood pH of equine endurance competitors are limited, then, discuss what has been done in other equine competitors or even what has been done in human marathon and/or triathlon competitors. The impact of this type of competition on the athlete, even outside of endurance and/or outside of the equine athlete, would be useful in understanding the value of the work being done within this study. In addition, much of the focus of the introduction and even within the discussion is on not finishing the endurance race within the equine athlete, but how is not finishing a race consider a bad thing? maybe the potential financial loss and/or the associated health risks that are involved with an athlete that cannot finish a race. Numbers of non-finishers does not explain why this research is important to those less familiar of the sport. Could not finishing an endurance race be associated with the rider not able to finish or behavioral issues with the horse that makes it hard to finish rather than reasons associated with the variables measured within this study? Could additional reasons include health problems not tested in this study such as lameness? Maybe research within the introduction supporting that non-finishing in endurance is primarily due to what was measured within this study would help to emphasize the value of this research. It would also be helpful to address within the introduction how alterations with hydro electrolyte and hematocrit parameters within the equine athlete can have a long-term negative impact on these animals. If research is lacking, a discussion even within the human athlete could be utilized to relate to potential issues within the equine athlete.

Au: The introduction was completely changed trying to make it as short as possible following the reviewer's recommendation. We fully agree with the reviewer that when a horse fails to finish a race there are several aspects to consider: the animal welfare (even if horses were healthy at the start of the race, both metabolic and locomotor problems could be the cause of not finishing the race; also, the economic aspects, important competition expenses are generated plus expenses in treatments will be added. It is true that some horses may be among the unfinished due to their rider; although they are not quantified, their occurrence is rare, but it is a piece of information that we are missing and that we could add in future works. In our study, we could not collect the causes of elimination, one limitation is that many horses are not presented to the veterinary check-up and we do not really know the cause of elimination. It would be interesting to be able to investigate the negative impact of dehydration in these horses, but we only have the test values. In a future trial and considering a smaller number of horses, we could monitor these more dehydrated horses during the races.

We have now included in the first paragraph of the discussion section the following phrase that is suggested by the reviewer that may help the reader to be conscious of the limitation of our study: “This is a descriptive analysis performed under field conditions where many variables as sample population and variability associated with horse background and the environment of the races could not be controlled. This limitation should be considered for the interpretation of the results.” Moreover, we have included another specific phrase that reads: “One of the limitations of the present study is the lack of information regarding the cause of elimination of the horse from the race.

We have also included a phrase in order to better explain the relevance/concept of not finish the race as the reviewer suggested: “Although RAID is a competitive sport, the primary goal for most competitors is the completion of the race; the main causes of horse retirement are perception by the rider of fatigue or lameness in the horse.”

Along with justifying what was studied and why it was studied, a hypothesis should be given after the objective statement within the introduction. In addition, while the authors add in a justification statement after the objective statement by saying that this research will “contribute to animal welfare and disease prevention”, these areas were not measured within this study. Welfare is usually measured within more behavioral based measurements using such markers as cortisol and vital signs, but this was not done within this study. In addition, no disease detection was given within this study and authors do not explain how disease relates to the measurements given. Authors need to stay focused on what was truly measured and it’s direct impact on the equine endurance athlete when including a justification.

Au: The following sentence was removed from the introduction: “Studying the pathophysiology of the hydro electrolyte imbalance in equine athletes will contribute to animal welfare and disease prevention”. We fully agree we cannot make statements about animal welfare since variables such as cortisol or other stress indicators were not taken. Besides the hypothesis has been included:  Our hypothesis is that the completion of the RAIDs and the hydroelectrolytic and acid base parameters after the race are associated with the comfort index.  We specially thank the reviewer for this criticism and suggestion.

As for the methods section, more details need to be given concerning the 900 horses sampled. Did each horse only do one race or was there a potential that a horse was sampled for more than one race for this study? Age, height, weight, breed, gender, amount of training, and which type of race attempted needs to be given for the horses sampled for this study as these variables can impact results. These variables can be reported utilizing means (SD) or ranges for the 900. This information would be useful given in a table format to visualize the differences between the three types of race distances reported. Furthermore, for determining the health of the horses utilized for this study, how thorough of a clinical examination was done for each horse? Did it include diagnostic testing utilizing radiographs, endoscopes, and ultrasound? Was a clinical history taken to determine any previous health issues tracked by a veterinarian? Was a full lameness evaluation done with flexion tests utilizing nerve blocks and hoof testers? Did the same veterinarian perform the health evaluations for all horses? How soon before the race were these health evaluations performed? What examinations were done during the race and was it done by the same veterinarian as the initial examination? How soon after was the blood samples taken at the end of the race and was it consistent for all?

Au: Some horses participated in several races, but it was taken as different horse. Unfortunately, we do not have the record of age, sex, race, type and time of training, and other important data, they are not recorded in the competition; but without a doubt that is a great lack that we will take into account in the following work. The pre-competition exam is an exam where clinical parameters such as heart rate, respiratory rate, hydration status, intestinal motility, capillary refill time are checked, and a locomotor control is also done, and it is examined if there is any claudication or not. Check-up are carried out the day before the race between 11 a.m. and 1 p.m. (approximately 20 hours before the race) by a team of veterinarians that will be the same until the end of the race. The veterinarians who work on the tests are part of a pool of veterinarians certified by the FEU. During the test, at the break, the veterinarians carry out the same metabolic and locomotor examination as the previous day, to control the health status of the horses and authorize those with parameters close to normal to continue in the race. In horses that are eliminated from the race at the break or that abandon the race, blood samples are taken immediately to measure hematocrit and plasma proteins. In horses that finish the race, blood samples are taken when the competition ends.

As for the discussion, this section begins with a focus on whether horses finished the race, but it does not appear that any documentation was made as to why the horses did not finish the race. Which horses were specifically pulled by the veterinarian inspecting the horses during the race, and of those pulled by the veterinarian, what specifically was the reasons documented for why? What was noted on their records such as an increased vital sign or signs of dehydration? While alterations may have been noted in the measurements recorded for this study for these animals, those alterations may not have stopped a competitor from finishing the race or may not have been the sole reason why. Without notes from the veterinarian for reasons why a horse was pulled or even from the owner, it is hard to support that the measurements taken by the authors were directly associated to not finishing the race. It is especially difficult to make any conclusions without controls utilized for the study. This would help to eliminate potential unforeseen variables that were not accounted for. This lack of a control along with an unbalanced sample population and variability associated with the horse background and with the environment of the races should be discussed as potential limitations to the study. The discussion section must include limitations to a study so that future studies may work to eliminate these potential issues.

Au: Totally agree with all of this. As we have previously mentioned, we have added two phrases in the discussion section commenting on the limitations of our study. But we will keep it in mind for our next job to record all this important information and improve the analysis. Even with these limitations, we still think that the study is novel and solid as a prospective study.

Round 2

Reviewer 2 Report (New Reviewer)

Still very concerned by the lack of control group or at a minimum, two samples taken before exercise.

Author Response

            February 3rd, 2023

Dear Editor

We want to thank the reviewers again for their important work to improve this manuscript.

The present manuscript was checked for English review according to the journal's recommendations (#59458, see the attached receipt of the payment); thus, the text may have changed slightly. We have included some phrases in the materials and methods and discussion section (originally included) to achieve the requirement of minimum 4000 words.

A small change was made to the title, according to one reviewer's recommendations. This change, in addition to notifying all reviewers, was notified to Mr. Sabin Mihai Bana, Section Managing Editor, to avoid future confusion.

Regarding the last comments of the reviewers:

REV # 2

“Still very concerned by the lack of control group or at a minimum, two samples taken before exercise.”

Authors: As exchanged in our previous response, this field study with 900 horses in RAIDs have to be subjected to what was possible to be performed. Unfortunately, the inclusion of a control group is not always easy in field trials during a competition, and the use of two basal samples (at two distant moments) is not easy to be accepted by the organizers and competitors. Thus, only one blood sample taken before (24 hours before) the race was what we could perform. This and other limitations of this study were exposed within the discussion for the correct interpretation of the findings by the readers of the article. On the other hand, we are taking this point into account in future research will fervently attempt to conduct controlled trials, and strengthen the baseline samples.

REV # 3

“The only suggested revision, thus, at this time, would be concerning the methods section as the authors gave some helpful information concerning the veterinary evaluations within the races when responding to reviewer comments. Readers would find this information useful in understanding the horses utilized within this study and the process of these types of races, and thus, this information needs to be included. Below are the authors' responses that need to be integrated within the methods section”

Authors: The full paragraph suggested by the reviewer was included in M&M within the subsections Competences and animals and Blood sampling.

Kind regards

The authors

Reviewer 3 Report (New Reviewer)

While some of the revisions suggested in the original review could not be addressed without redoing the study, the authors did a thorough job of revising as much as they could do at this time appreciating the recommended revisions given. It is recommended to the authors that they consider for future studies the suggested revisions and suggestions given in the original review.

The only suggested revision, thus, at this time, would be concerning the methods section as the authors gave some helpful information concerning the veterinary evaluations within the races when responding to reviewer comments. Readers would find this information useful in understanding the horses utilized within this study and the process of these types of races, and thus, this information needs to be included. Below are the authors' responses that need to be integrated within the methods section:

"The pre-competition exam is an exam where clinical parameters such as heart rate, respiratory rate, hydration status, intestinal motility, capillary refill time are checked, and a locomotor control is also done, and it is examined if there is any claudication or not. Check-up are carried out the day before the race between 11 a.m. and 1 p.m. (approximately 20 hours before the race) by a team of veterinarians that will be the same until the end of the race. The veterinarians who work on the tests are part of a pool of veterinarians certified by the FEU. During the test, at the break, the veterinarians carry out the same metabolic and locomotor examination as the previous day, to control the health status of the horses and authorize those with parameters close to normal to continue in the race. In horses that are eliminated from the race at the break or that abandon the race, blood samples are taken immediately to measure hematocrit and plasma proteins. In horses that finish the race, blood samples are taken when the competition ends."

Author Response

We would like to thank the reviewer for the important work to improve this manuscript.

The present manuscript was checked for English review according to the journal's recommendations. So it may appear slightly different. A small change was made to the title, according to one reviewer's recommendations. This change, in addition to notifying all reviewers, was notified to Mr. Sabin Mihai Bana, Section Managing Editor, in order to avoid future confusion.

REPLY

The only suggested revision, thus, at this time, would be concerning the methods section as the authors gave some helpful information concerning the veterinary evaluations within the races when responding to reviewer comments. Readers would find this information useful in understanding the horses utilized within this study and the process of these types of races, and thus, this information needs to be included. Below are the authors' responses that need to be integrated within the methods section”

The full paragraph suggested by the reviewer was included in M&M within the subsections Competences and animals and Blood sampling.

Kind regards

The authors

This manuscript is a resubmission of an earlier submission. The following is a list of the peer review reports and author responses from that submission.

Round 1

Reviewer 1 Report

I appreciate all the work that went into this study. However, the authors can need to improve some aspects.

- Line 102: viniculture should be replace for venipuncture

- Line 308 to 310: how can explain the authors this results? Supposedly horses in CI high should have higher pH because the losses of chloride but in the study horses in CI high had lower pH. Can you explain these results and add the comments in the discussion?

- Lines 333 to 335. The sentence should rewriter.